# Regular Fourier Features
# for Nonstationary Gaussian Processes

## Abstract

Simulating a Gaussian process requires sampling from a high-dimensional Gaussian distribution, which scales cubically with the number of sample locations. Spectral methods address this challenge by exploiting the Fourier representation and treating the spectral density as a probability distribution suitable for Monte Carlo approximation. Although this probabilistic interpretation is valid for stationary processes, it is overly restrictive for the nonstationary case, where spectral densities are generally not probability measures. We propose regular Fourier features for harmonizable processes to avoid this limitation. Our method discretizes the spectral representation directly, preserving the correlation structure among spectral weights without requiring probability assumptions. Assuming finite spectral support, this yields an efficient low-rank approximation that is positive semi-definite by construction and consistent under mild regularity conditions. When the spectral density is unknown, the framework also extends to kernel learning from data, which we explore as a proof of concept. We demonstrate the approximation on locally stationary and harmonizable mixture kernels, the latter with a complex-valued spectral density. As a feasibility study, we then apply the kernel-learning extension to real and synthetic data, where it matches competitive baselines.

## 1 Introduction

Gaussian processes (GPs) are widely used in fields such as machine learning (Rasmussen & Williams, 2005) and geostatistics (Higdon et al., 1998). Their appeal lies in flexible nonparametric regression with principled uncertainty estimates and closed-form computations. These closed-form solutions are powerful but computationally expensive: simulating a GP at $n$ locations requires decomposing an $n \times n$ kernel matrix, which scales as $\mathcal{O}(n^3)$ (Rasmussen & Williams, 2005).

Spectral methods offer an efficient alternative by exploiting the Fourier representation of stochastic processes. For stationary GPs, Rahimi & Recht (2007) introduced random Fourier features (RFFs), which treat the spectral density as a probability measure and approximate the kernel via Monte Carlo sampling. This yields a low-rank approximation with complexity $\mathcal{O}(nm^2)$, where $m \ll n$ is the number of features.

Extending RFFs to nonstationary processes requires treating $s(\omega, \omega')$ as a probability measure, an assumption that generally fails to hold. Existing spectral extensions (Samo & Roberts, 2015; Ton et al., 2018) make this assumption, which limits the class of representable kernels. In addition, their approximation is typically a biased and inconsistent estimator of the true kernel.

We propose regular Fourier features for one-dimensional harmonizable GPs that avoid these limitations. Our approach discretizes the spectral representation on a regular grid, following the classical Riemann sum approximation of Shinozuka & Jan (1972) for stationary simulation. For nonstationary processes, the spectral weights at different frequencies are correlated according to the spectral density $s(\omega_i, \omega_j)$. Preserving this correlation structure yields a low-rank approximation that is positive semi-definite by construction.

Our contributions are:

- We extend regular Fourier features to harmonizable processes without a probability interpretation; the resulting low-rank approximation is consistent (Theorem 2) and handles complex-valued spectral densities.

- We introduce a factorized spectral parametrization that guarantees a valid nonstationary kernel by construction and enables learning from data.

## 2 Background

We consider harmonizable GPs, a broad class that includes stationary processes as a special case and encompasses many nonstationary processes of practical interest (Yaglom, 1987; Shen et al., 2019). A zero-mean GP is called harmonizable when it admits the spectral representation (Cramér, 1942; 1946; Loève, 1948)

$$Z(x) = \int_{\mathbb{R}} \exp(i\omega x)\, d\Gamma(\omega), \tag{1}$$

where $\{\Gamma(\omega)\}$ is a complex-valued zero-mean stochastic process indexed by $\omega$, whose spectral distribution

$$S(\omega, \omega') = \mathrm{E}[\Gamma(\omega)\overline{\Gamma(\omega')}], \tag{2}$$

has bounded variation on $\mathbb{R} \times \mathbb{R}$ (Loève, 1948). The spectral distribution $S$ is positive semi-definite and determines a positive semi-definite kernel $k(x, x') = \mathrm{E}[Z(x)\overline{Z(x')}]$ via the Fourier–Stieltjes integral (Loève, 1948)

$$k(x, x') = \iint_{\mathbb{R}^2} \exp(i(\omega x - \omega' x'))\, dd' S(\omega, \omega'). \tag{3}$$

If the spectral distribution is differentiable, it admits a spectral density $s(\omega, \omega')$, and the integral reduces to

$$k(x, x') = \iint_{\mathbb{R}^2} \exp(i(\omega x - \omega' x'))\, s(\omega, \omega')\, d\omega\, d\omega'. \tag{4}$$

If $\{Z(x)\}$ is stationary, the spectral increments at distinct frequencies are uncorrelated, so the spectral measure concentrates on the diagonal $\omega = \omega'$. The kernel then depends only on the lag, and the double integral collapses to

$$k(x - x') = \int_{\mathbb{R}} \exp(i\omega(x - x'))\, dS(\omega), \tag{5}$$

or, when a spectral density $s(\omega)$ exists,

$$k(x - x') = \int_{\mathbb{R}} \exp(i\omega(x - x'))\, s(\omega)\, d\omega. \tag{6}$$

### 2.1 Random Fourier Features

For stationary processes, Rahimi & Recht (2007) observed that the spectral density $s(\omega)$, up to a scaling constant $\sigma^2 = k(0)$, is a probability density. This allows Monte Carlo approximation of the kernel

$$k(x - x') \approx \frac{\sigma^2}{m} \sum_{j=1}^{m} \exp(i\Omega_j x) \exp(-i\Omega_j x')$$

$$= \langle \boldsymbol{\Phi}(x), \boldsymbol{\Phi}(x') \rangle, \tag{7}$$

where $\Omega_j \sim p(\omega) \propto s(\omega)$, $\langle \boldsymbol{a}, \boldsymbol{b} \rangle = \sum_k a_k \bar{b}_k$ denotes the inner product, and the feature map is

$$\boldsymbol{\Phi}(x) = \frac{\sigma}{\sqrt{m}} \left( \exp(i\Omega_1 x) \quad \dots \quad \exp(i\Omega_m x) \right). \tag{8}$$

Table 1: Comparison of Fourier-feature approximation methods for GPs. Checkmarks indicate supported features: nonstationary kernels, consistent approximation, positive semi-definiteness at finite rank, unconstrained spectral densities (no probability or real-valuedness assumption), and unbounded spectral support. Our method accepts bounded support in exchange for the remaining four features. The Samo & Roberts (2015, Th. 7) row refers to their universal density theorem; its Monte Carlo form has the same limitations as Ton et al. (2018).

| Method | Nonstationary $k$ | Consistent | Finite-rank PSD | Unconstrained $s$ | Unbounded $s$ |
|---|---|---|---|---|---|
| Rahimi & Recht (2007) | ✗ | ✓ | – | – | ✓ |
| Samo & Roberts (2015, Th. 7) | ✓ | ✓ | ✗ | ✗ | ✓ |
| Ton et al. (2018) | ✓ | ✗ | ✓ | ✗ | ✓ |
| This work | ✓ | ✓ | ✓ | ✓ | ✗ |

The low-rank form $\langle \boldsymbol{\Phi}(x), \boldsymbol{\Phi}(x') \rangle$ ensures positive semi-definiteness by construction.

Generalizations to nonstationary processes (Samo & Roberts, 2015; Ton et al., 2018) require a probability interpretation of $s(\omega, \omega')$. A naïve Monte Carlo approximation would be

$$k(x, x') \approx \frac{\sigma^2}{m} \sum_{j=1}^{m} \exp(\mathrm{i}(\Omega_j x - \Omega'_j x')), \tag{9}$$

where $(\Omega_j, \Omega'_j) \sim p(\omega, \omega') \propto s(\omega, \omega')$. This approximation is not guaranteed to be positive semi-definite since $\Omega_j \neq \Omega'_j$ in general. This becomes clear by setting $m = 1$ and $x = x'$:

$$k(x, x) \approx \sigma^2 \exp(\mathrm{i}(\Omega_1 - \Omega'_1)x), \tag{10}$$

which is not guaranteed to be real, let alone nonnegative.

To address this, Samo & Roberts (2015); Ton et al. (2018) construct modified feature maps that ensure positive semi-definite approximations, but still constrain the spectral density to a probability measure. Consequently, kernel approximation or kernel learning is restricted to this class, and the low-rank approximation is biased and inconsistent (see Appendix A). For Samo & Roberts (2015), the inconsistency arises only in their Monte Carlo form: their universal density theorem is consistent, but its approximations are generally not positive semi-definite.

In contrast, our approach (Section 3) discretizes the spectral representation on a regular grid and preserves the correlations among spectral increments. This removes the probability-measure restriction and yields an approximation that is positive semi-definite by construction and consistent (Theorem 2). Table 1 contrasts it with the Fourier-feature methods above.

## 2.2 Related Work

Bach (2017) showed that kernel quadrature rules are a special case of random feature expansions, writing the kernel as an expectation $k(x, x') = \mathrm{E}_\Omega[\varphi(\Omega, x)\varphi(\Omega, x')]$ of one-dimensional random features. Quadrature rules improve sample efficiency by replacing naïve Monte Carlo sampling with a weighted sum $k(x, x') \approx \sum_j a_j \varphi(\omega_j, x)\varphi(\omega_j, x')$: Dao et al. (2017) fix $\varphi(\omega, x) = \exp(\mathrm{i}\omega x)$ and vary the spectral density, which restricts the approach to stationary kernels. Munkhoeva et al. (2018) instead fix a Gaussian distribution and vary $\varphi(\omega, x)$, reaching some nonstationary kernels but only those expressible as such an expectation under the fixed distribution. Our regular grid with Riemann-sum weights is the simplest quadrature rule, but we apply it to a different object: the Fourier–Stieltjes integral (4). In this integral, the integrand couples two frequencies, and $s(\omega, \omega')$ need not be a probability distribution; positive semi-definiteness holds not term by term, as in the quadrature approaches, but through the preserved correlations among the spectral increments.

A complementary line of work approximates inference rather than the model. Sparse variational GPs (Titsias, 2009; Hensman et al., 2013) represent the posterior through inducing points and leave the kernel unchanged. Variational Fourier features (Hensman et al., 2018) use projections of the process onto harmonics on a

bounded interval as their inducing variables, although these projections are available in closed form only for Matérn kernels. They place the harmonics on a regular frequency grid and use the grid to approximate the posterior, whereas we use ours to approximate the harmonizable process. Unlike a stationary GP, a harmonizable GP with an integrable kernel has a well-defined Fourier transform, and Shen et al. (2019) evaluate it directly at learned frequencies.

Finally, several methods fix a parametric family of spectral densities. Wilson & Adams (2013) model the stationary spectral density as a Gaussian mixture, while Benton et al. (2019) model it nonparametrically with a GP. Remes et al. (2017) extend the mixture idea to nonstationary kernels with a real-valued spectral density, a subclass of the harmonizable family. They further let frequencies, length scales, and amplitudes vary with the input (Remes et al., 2017; 2018), although the explicit spectral density of the resulting kernel remains unknown. Shen et al. (2019) model the nonstationary kernel as a mixture of locally stationary kernels (LSKs) (Silverman, 1957), a family dense in the harmonizable class that, in particular, admits complex-valued spectral densities. These methods specify the kernel itself rather than a low-rank approximation of it.

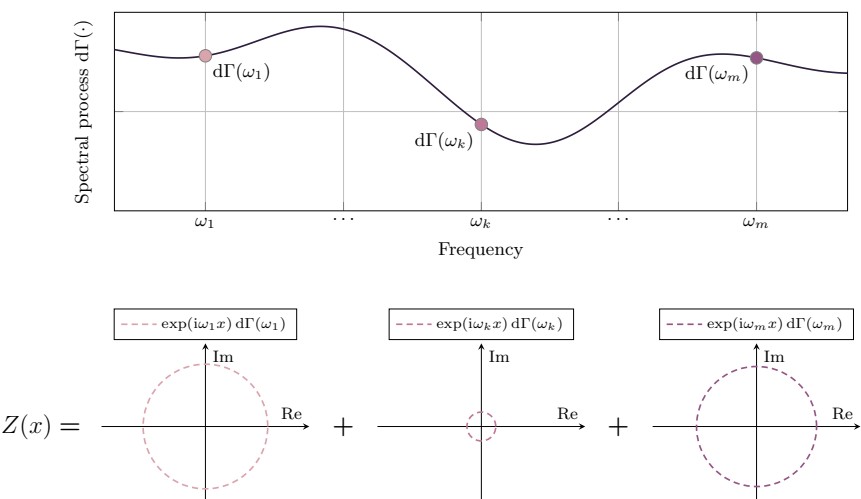

Figure 1: Regular Fourier features approximate the stochastic process $Z(x)$ as a sum of complex exponentials weighted by correlated spectral increments $\mathrm{d}\Gamma(\omega_k)$. The spectral process (top) is discretized at frequencies $\omega_1, \ldots, \omega_m$. Each harmonic (bottom) shows the contribution $\mathrm{d}\Gamma(\omega_k)\exp(\mathrm{i}\omega_k x)$ in the complex plane.

## 3 Nonstationary Regular Fourier Features

We derive regular Fourier features for harmonizable processes by discretizing the spectral representation on an equidistant frequency grid, following the classical simulation idea of Shinozuka & Jan (1972). While the Riemann sum approximation itself is standard, its extension to harmonizable GPs has not been explored. Unlike random approaches that require a probability interpretation, our method preserves the correlation structure among spectral weights without modifying the spectral density.

The spectral representation (1) is a Riemann–Stieltjes integral that can be approximated by a finite sum (Loève, 1978) (see Figure 1) on a symmetric, equidistant frequency grid with spacing $\Delta\omega = \omega_{k+1} - \omega_k$

$$Z(x) \approx \sum_{k=-m}^{m-1} \exp(\mathrm{i}\omega_k x) W_k, \tag{11}$$

where $W_k := \Gamma(\omega_k + \Delta\omega) - \Gamma(\omega_k)$ is the spectral increment at frequency $\omega_k$. The random variables $W = (W_{-m}, \ldots, W_{m-1})^\top$ form a zero-mean complex-valued random vector with covariance

$$\mathrm{E}[WW^\dagger] \approx \boldsymbol{S}\Delta\omega^2, \quad [\boldsymbol{S}]_{ij} = s(\omega_i, \omega_j). \tag{12}$$

Here $(\cdot)^{\dagger}$ denotes the conjugate transpose. In the nonstationary case, the spectral weights $\{W_k\}$ are no longer independent but are correlated according to the spectral density $s(\omega_i, \omega_j)$. This off-diagonal correlation structure encodes the nonstationarity of the process.

When $W$ is complex-valued, its distribution is not fully determined by $\mathrm{E}[WW^{\dagger}]$ alone; the pseudo-covariance $\mathrm{E}[WW^{\top}]$ is also required (Lee & Messerschmitt, 1994). We assume $W$ is circular ($\mathrm{E}[WW^{\top}] = 0$) and Gaussian, so that $\mathrm{E}[WW^{\dagger}] \approx \boldsymbol{S}\Delta\omega^2$ fully specifies the distribution. This assumption does not restrict the class of harmonizable GPs that our method can represent (see Appendix C). Factoring $\boldsymbol{S}\Delta\omega^2 = \boldsymbol{C}\boldsymbol{C}^{\dagger}$, we write

$$Z(x) \approx \boldsymbol{\alpha}(x)\boldsymbol{C}E = \boldsymbol{\varphi}(x)E, \tag{13}$$

where the row vector $[\boldsymbol{\alpha}(x)]_k = \exp(\mathrm{i}\omega_k x)$ collects the Fourier basis functions, $\boldsymbol{\varphi}(x) = \boldsymbol{\alpha}(x)\boldsymbol{C}$ is the feature map, and $E = E_{\mathrm{re}} + \mathrm{i}E_{\mathrm{im}}$ is a circularly symmetric complex Gaussian with $E_{\mathrm{re}}, E_{\mathrm{im}} \sim \mathcal{N}(\boldsymbol{0}, \frac{1}{2}\boldsymbol{I})$.

### 3.1 Low-Rank Kernel Approximation

The kernel $k(x, x') = \mathrm{E}[Z(x)\overline{Z(x')}]$ admits the low-rank approximation

$$k(x, x') \approx \langle \boldsymbol{\varphi}(x), \boldsymbol{\varphi}(x') \rangle = \boldsymbol{\alpha}(x)\boldsymbol{C}\boldsymbol{C}^{\dagger}\boldsymbol{\alpha}(x')^{\dagger}, \tag{14}$$

which is positive semi-definite by construction since the resulting kernel matrix has the form $\boldsymbol{\Phi}\boldsymbol{\Phi}^{\dagger}$ for any feature matrix $\boldsymbol{\Phi}$. This contrasts with the naïve Monte Carlo approximation (9), which lacks this guarantee.

For $n$ sample locations, the kernel matrix $\boldsymbol{K} \in \mathbb{R}^{n \times n}$ is approximated as

$$\boldsymbol{K} \approx \boldsymbol{L}\boldsymbol{L}^{\dagger}, \quad [\boldsymbol{L}]_i = \boldsymbol{\varphi}(x_i), \quad \boldsymbol{L} \in \mathbb{C}^{n \times 2m}. \tag{15}$$

**Real-Valued Processes** For real-valued processes, the spectral weights satisfy Hermitian symmetry $W_{-k} = \overline{W}_k$. This symmetry is incompatible with circularity, since $\mathrm{E}[Z(x)Z(x')] = k(x, x') \neq 0$. We therefore sample circular weights $W = \boldsymbol{C}E$ as in Eq. (13) and symmetrize the expansion

$$Z(x) \approx \frac{\sqrt{2}}{2} \sum_{k=-m}^{m-1} \left( \exp(\mathrm{i}\omega_k x)W_k + \exp(-\mathrm{i}\omega_k x)\overline{W}_k \right) = \sqrt{2}\,\mathrm{Re}[\boldsymbol{\alpha}(x)W], \tag{16}$$

where the row vector $[\boldsymbol{\alpha}(x)]_k = \exp(\mathrm{i}\omega_k x)$.

The factor $\sqrt{2}$ arises because for circular $W$ the real part carries only half the power of the complex process (13): $\mathrm{E}[\mathrm{Re}[\boldsymbol{\alpha}(x)W]\,\mathrm{Re}[\boldsymbol{\alpha}(x')W]] = \frac{1}{2}\,\mathrm{Re}[\langle\boldsymbol{\varphi}(x), \boldsymbol{\varphi}(x')\rangle]$. Matching the second-order structure of the complex process fixes the scale to $\sqrt{2}$. The kernel approximation is then deterministic and has the form

$$k(x, x') \approx \mathrm{Re}[\langle\boldsymbol{\varphi}(x), \boldsymbol{\varphi}(x')\rangle], \quad \boldsymbol{\varphi}(x) = \boldsymbol{\alpha}(x)\boldsymbol{C}. \tag{17}$$

The low-rank approximation of the kernel matrix is then $\boldsymbol{K} \approx \boldsymbol{L}\boldsymbol{L}^{\top}$ with $[\boldsymbol{L}]_i = (\mathrm{Re}[\boldsymbol{\varphi}(x_i)], \mathrm{Im}[\boldsymbol{\varphi}(x_i)])$ and $\boldsymbol{L} \in \mathbb{R}^{n \times 4m}$, hence rank at most $4m$.

When the spectral weights $W_k$ are additionally real-valued ($W_{-k} = \overline{W}_k = W_k$), no circularity assumption is required, and the positive- and negative-frequency terms combine to $2\cos(\omega_k x)W_k$. The feature map then simplifies to $[\boldsymbol{\alpha}(x)]_k = \cos(\omega_k x)$ for $\omega_k \neq 0$ and $1/2$ otherwise, yielding the kernel $k(x, x') \approx 4\,\boldsymbol{\alpha}(x)\boldsymbol{C}\boldsymbol{C}^{\top}\boldsymbol{\alpha}(x')^{\top}$. For real-valued spectral weights, we use a one-sided grid of $m$ nodes, so $W$ and $\boldsymbol{C}$ have dimension $m$ or $m \times m$, and the low-rank approximation has rank at most $m$.

**Assumptions and Limitations** The method assumes finite spectral support: $s(\omega, \omega') = 0$ for $|\omega| > \omega_m$ or $|\omega'| > \omega_m$. This band-limited assumption is natural in many applications. When data are sampled at intervals $\Delta x$, the Nyquist theorem limits recoverable frequencies to $\omega_{\mathrm{Nyquist}} = \pi/\Delta x$. Spectral content beyond this is aliased and fundamentally unidentifiable from observations. By setting $\omega_m < \omega_{\mathrm{Nyquist}}$, our method avoids modeling high-frequency components that cannot be distinguished from the data. This is

particularly relevant in surface texture measurement, where the textures are inherently band-limited (ISO 21920-2, 2021; ISO 25178-2, 2021). When the spectral support is not strictly finite, the method introduces a truncation error proportional to the spectral mass beyond $\omega_m$ (see Theorem 2).

The finite sum (11) is periodic with period $T = 2\pi/\Delta\omega$. Thus, our low-rank approximation estimates $\sum_{p,q} k(x - pT, x' - qT)$, introducing aliasing error. If the kernel is not zero for $|x|, |x'| > \pi/\Delta\omega$, the periods interfere with the approximation. In practice, $\Delta\omega$ should be chosen such that $x_{\max} < \pi/\Delta\omega$, where $x_{\max} = \max_i |x_i|$.

Numerically, the Cholesky factorization $\boldsymbol{S}\Delta\omega^2 = \boldsymbol{C}\boldsymbol{C}^\dagger$ requires $\boldsymbol{S}\Delta\omega^2$ to be positive definite, whereas it is in general only positive semi-definite. Adding a small jitter $\epsilon$ on the diagonal restores positive definiteness but introduces a band-limited white-noise error, confined to the modeled band $[-\omega_m, \omega_m)$. This contrasts with the same jitter applied to the standard kernel matrix, which introduces full-band white noise. As $\epsilon$ is usually small, the error is negligible.

Finally, scaling to high dimensions is the central challenge. A general, non-separable $d$-dimensional harmonizable kernel requires discretizing the joint spectral density $s(\boldsymbol{\omega}, \boldsymbol{\omega}')$ on a $2d$-dimensional frequency grid; the resulting spectral covariance matrix has on the order of $m^{2d}$ entries, so both storage and factorization grow exponentially in $d$ and the computational gain of the low-rank approximation is lost. Product kernels $k(\boldsymbol{x}, \boldsymbol{x}') = \prod_{p=1}^{d} k(x_p, x'_p)$ keep the $\mathcal{O}(ndm^2)$ cost but impose separability. Another option is adaptive placement of frequency nodes where the spectral density has mass. A general solution is left to future work. Accordingly, this work addresses only one-dimensional kernels.

## 3.2 Approximation Error

The truncation error and the discretization error (including aliasing) can be quantified. On the symmetric grid $\omega_k = k\Delta\omega$, $k = -m, \ldots, m-1$, with cutoff $\omega_m = m\Delta\omega$, the low-rank approximation (14) of the kernel reads

$$\hat{k}(x, x') = \sum_{i,j=-m}^{m-1} \exp(\mathrm{i}(\omega_i x - \omega_j x'))\, s(\omega_i, \omega_j)\, \Delta\omega^2, \tag{18}$$

and $\mathcal{I}_m := [-\omega_m, \omega_m)^2$ denotes the square covered by the frequency grid. The following lemma gives a mild regularity condition on the spectral distribution $S(\omega, \omega')$ under which the spectral density $s(\omega, \omega')$ exists and is well-behaved; it also justifies the covariance structure (12) underlying our model. The theorem then bounds the approximation error under the same condition. Both are proved in Appendix B.

**Lemma 1.** *Let $\{Z(x)\}$ be a harmonizable stochastic process whose spectral distribution is $S(\omega, \omega')$, and suppose the second generalized derivative of $S(\omega, \omega')$ exists, is finite, and is continuous at every diagonal point $(\omega, \omega)$, $\omega \in \mathbb{R}$. Then the spectral density $s(\omega, \omega') := \partial^2 S/\partial\omega\,\partial\omega'$ exists, is finite and continuous on $\mathbb{R} \times \mathbb{R}$, and is the covariance of the quadratic mean (q.m.) derivative process $\{\Gamma'(\omega)\}$,*

$$s(\omega, \omega') = \mathrm{E}[\Gamma'(\omega)\overline{\Gamma'(\omega')}].$$

*Moreover, $\{\Gamma'(\omega)\}$ is q.m. continuous and $s(\omega, \omega')$ is absolutely integrable, $\iint_{\mathbb{R}^2} |s(\omega, \omega')|\, \mathrm{d}\omega\mathrm{d}\omega' \leq c < \infty$.*

**Theorem 2.** *Under the hypotheses of Lemma 1, let $s(\omega, \omega')$ and $c$ be as therein and let $\rho_m(\cdot)$ be the modulus of continuity of $s(\omega, \omega')$ on the compact closure of $\mathcal{I}_m$. Then the following hold.*

*(i) Error bound. For all $x, x' \in \mathbb{R}$,*

$$|\hat{k}(x, x') - k(x, x')| \leq \underbrace{4\omega_m^2 \rho_m\big(\sqrt{2}\,\Delta\omega\big) + (|x| + |x'|)\,\Delta\omega\, c}_{\text{discretization}} + \underbrace{\iint_{\mathbb{R}^2 \setminus \mathcal{I}_m} |s(\omega, \omega')|\, \mathrm{d}\omega\mathrm{d}\omega'}_{\text{truncation}}.$$

*(ii) Uniform consistency. Let $\mathcal{X} \subset \mathbb{R}$ be bounded. Then, in the iterated limit,*

$$\lim_{\omega_m \to \infty} \lim_{\Delta\omega \to 0} \sup_{x,x' \in \mathcal{X}} |\hat{k}(x, x') - k(x, x')| = 0.$$

**Truncation and aliasing** If the support of $s(\omega, \omega')$ lies in $\mathcal{I}_m$, as assumed above, the truncation term vanishes and consistency follows from the single limit $\Delta\omega \to 0$ at fixed $\omega_m$. The error term $(|x| + |x'|)\,\Delta\omega\,c$ reflects the aliasing effect discussed above, since $\hat{k}(x, x')$ is $2\pi/\Delta\omega$-periodic. When there is no discretization $(\Delta\omega \to 0)$, there is no aliasing error.

### 3.3 Extension to Kernel Learning

When the spectral density is unknown, the framework extends to kernel learning. We focus on real-valued kernels throughout; note that the spectral density $s(\omega, \omega')$ can still be complex-valued. A valid spectral density must be positive semi-definite and fulfill

$$s(\omega, \omega') = \overline{s(\omega', \omega)} = \overline{s(-\omega, -\omega')}. \tag{19}$$

We parametrize the spectral density as

$$s(\omega, \omega') = \langle \boldsymbol{f}(\omega'), \boldsymbol{f}(\omega) \rangle + \langle \boldsymbol{f}(-\omega), \boldsymbol{f}(-\omega') \rangle, \tag{20}$$

which is positive semi-definite and Hermitian by construction. The second term enforces the real-kernel symmetry $s(\omega, \omega') = \overline{s(-\omega, -\omega')}$. Both properties hold for arbitrary $\boldsymbol{f}\colon \mathbb{R} \to \mathbb{C}^r$. In matrix form, the spectral matrix is

$$\boldsymbol{S} = \boldsymbol{F}\boldsymbol{F}^\dagger + \boldsymbol{F}_-\boldsymbol{F}_-^\dagger, \tag{21}$$

where $[\boldsymbol{F}]_{kj} = \overline{f_j(\omega_k)}$ and $[\boldsymbol{F}_-]_{kj} = f_j(-\omega_k)$, both in $\mathbb{C}^{2m\times r}$. The function $\boldsymbol{f}$ can be parametrized by a neural network with $2r$ real outputs representing the real and imaginary parts.

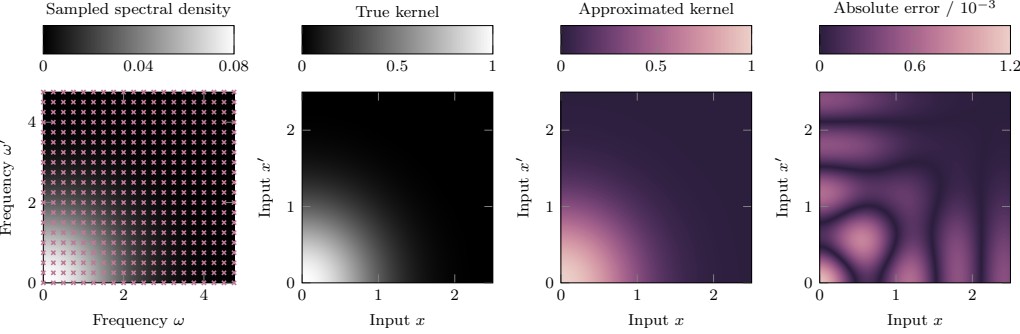

Figure 2: Low-rank approximation of the LSK showing the sampled spectral density, true kernel, approximation, and absolute error.

**Inference** Given $n$ observations $\{(x_i, z_i)\}_{i=1}^n$ and a Gaussian likelihood, we maximize an objective such as the marginal log-likelihood

$$\log p(\boldsymbol{z} \mid \Theta) \propto -\boldsymbol{z}^\top \boldsymbol{\Sigma}^{-1}\boldsymbol{z} - \log|\boldsymbol{\Sigma}|, \tag{22}$$

where $\boldsymbol{z} = (z_1, \ldots, z_n)^\top$, $\boldsymbol{\Sigma} = \boldsymbol{K} + \sigma_{\text{noise}}^2\boldsymbol{I}$, and $\Theta$ includes the network parameters and noise variance $\sigma_{\text{noise}}^2$.

The factorized form avoids the $\mathcal{O}(m^3)$ matrix decomposition of $\boldsymbol{S}\Delta\omega^2$. Since $\boldsymbol{S}\Delta\omega^2 = \boldsymbol{F}\Delta\omega(\boldsymbol{F}\Delta\omega)^\dagger + \boldsymbol{F}_-\Delta\omega(\boldsymbol{F}_-\Delta\omega)^\dagger$, the decomposition is

$$\boldsymbol{C} = \begin{pmatrix} \boldsymbol{F} & \boldsymbol{F}_- \end{pmatrix}\Delta\omega \in \mathbb{C}^{2m\times 2r}. \tag{23}$$

Then the real-valued kernel is $\boldsymbol{K} \approx \mathrm{Re}[(\boldsymbol{\Phi}\boldsymbol{C})(\boldsymbol{\Phi}\boldsymbol{C})^\dagger]$, where $[\boldsymbol{\Phi}]_{ik} = [\boldsymbol{\alpha}(x_i)]_k$ as in (17). Writing $\boldsymbol{\Phi}\boldsymbol{C} = \boldsymbol{A} + \mathrm{i}\boldsymbol{B}$ with real and imaginary parts in $\mathbb{R}^{n\times 2r}$, we obtain $\boldsymbol{K} \approx \boldsymbol{L}\boldsymbol{L}^\top$ with

$$\boldsymbol{L} = \begin{pmatrix} \boldsymbol{A} & \boldsymbol{B} \end{pmatrix} \in \mathbb{R}^{n\times 4r}, \tag{24}$$

which has rank at most $4r$. Using the Woodbury formula and matrix determinant lemma, the marginal log-likelihood costs $\mathcal{O}(nmr)$ when $r < m \ll n$.

**Posterior Prediction** Given $t$ test points, the posterior mean and covariance are

$$
\begin{aligned}
\boldsymbol{\mu}_{t|n} &= \boldsymbol{K}_{t,n}\boldsymbol{\Sigma}^{-1}\boldsymbol{z}, \\
\boldsymbol{\Sigma}_{t|n} &= \boldsymbol{K}_{t,t} - \boldsymbol{K}_{t,n}\boldsymbol{\Sigma}^{-1}\boldsymbol{K}_{n,t},
\end{aligned}
\tag{25}
$$

where $\boldsymbol{K}_{t,n} \in \mathbb{R}^{t \times n}$ is the kernel between test and training points. Using the low-rank form $\boldsymbol{K}_{t,n} \approx \boldsymbol{L}_* \boldsymbol{L}^\top$ with test feature matrix $\boldsymbol{L}_* \in \mathbb{R}^{t \times 4r}$, these simplify to $\boldsymbol{\mu}_* = \boldsymbol{L}_* \boldsymbol{\beta}$ and $\boldsymbol{\Sigma}_{t|n} = \boldsymbol{L}_* \boldsymbol{Q} \boldsymbol{L}_*^\top$. Here $\boldsymbol{\beta} = \boldsymbol{L}^\top \boldsymbol{\Sigma}^{-1}\boldsymbol{z} \in \mathbb{R}^{4r}$ and $\boldsymbol{Q} = \boldsymbol{I} - \boldsymbol{L}^\top \boldsymbol{\Sigma}^{-1}\boldsymbol{L} \in \mathbb{R}^{4r \times 4r}$ are cached during training. Prediction costs $\mathcal{O}(tmr)$, dominated by constructing $\boldsymbol{L}_*$.

## 4 Experiments

We evaluate our approach on two tasks: (1) low-rank kernel approximation, where the spectral density is known in closed form (Section 4.1); and (2) kernel learning, where the spectral density is learned from data (Section 4.2).

### 4.1 Low-Rank Kernel Approximation

We test our kernel approximation on two harmonizable kernels: the LSK (Silverman, 1957) and a harmonizable mixture kernel (HMK) (Shen et al., 2019). The LSK, with its real-valued spectral density, validates approximation quality on a case where existing RFF methods also apply. The HMK demonstrates a capability that existing RFF methods fundamentally cannot handle: approximating kernels with complex-valued spectral densities.

We measure approximation quality using the relative error

$$
\|\boldsymbol{K}_{\text{approx}} - \boldsymbol{K}_{\text{true}}\|_{\text{F}} / \|\boldsymbol{K}_{\text{true}}\|_{\text{F}},
\tag{26}
$$

where $\boldsymbol{K}_{\text{true}}$ is the true kernel matrix, $\boldsymbol{K}_{\text{approx}}$ its approximation, and $\|\cdot\|_{\text{F}}$ the Frobenius norm.

Table 2: Comparison of our method against the nonstationary RFF (Samo & Roberts, 2015; Ton et al., 2018). Relative error in % on the LSK ($a = 1$, $\Delta x = 10^{-3}$, $n = 2500$) versus $m$. The nonstationary RFF results are the mean with 95 % confidence intervals over 10 seeds.

| $m$ | 20 | 50 | 200 | 500 | 2000 |
|---|---|---|---|---|---|
| Ours | 0.102 | 0.085 | 0.077 | 0.076 | 0.076 |
| Nonstationary RFF | $107.2 \pm 13.4$ | $109.9 \pm 5.6$ | $105.8 \pm 4.4$ | $106.8 \pm 1.8$ | $107.1 \pm 1.6$ |

**Locally Stationary Kernel** The LSK is (Silverman, 1957)

$$
k_{\text{LSK}}(x, x') = \exp(-2a\bar{x}^2)\exp(-\frac{a}{2}\tilde{x}^2),
\tag{27}
$$

where $\bar{x} = (x + x')/2$ is the midpoint, $\tilde{x} = x - x'$ is the lag, and $a$ is a kernel parameter. This kernel is harmonizable with real-valued spectral density

$$
s_{\text{LSK}}(\omega, \omega') = \frac{1}{4\pi a}\exp(-\frac{1}{2a}\bar{\omega}^2)\exp(-\frac{1}{8a}\tilde{\omega}^2),
\tag{28}
$$

with $\bar{\omega} = (\omega + \omega')/2$ and $\tilde{\omega} = \omega - \omega'$. This spectral density has the additional property $s(\omega, \omega') = s(\omega, -\omega')$, so its spectral weights are real-valued: $\overline{W}_k = W_k$.

We set $a = 1.0$ and approximate the kernel on $x_i = i\Delta x$ for $i = 0, \ldots, n - 1$ with $\Delta x = 0.001$ and $n = 2500$. We use $m = 20$ on a one-sided frequency grid with cutoff frequency $\omega_m = 5$. The aliasing condition $\Delta\omega < \pi/(n\Delta x)$ is satisfied. Figure 2 shows (left to right): the spectral density $s_{\text{LSK}}(\omega, \omega')$ with the $m \times m$ sampling grid overlaid, the true kernel, our low-rank approximation, and the absolute error. The

approximation captures the kernel structure accurately, with absolute errors on the order of $10^{-3}$. Table 2 compares our method with the nonstationary RFF (Samo & Roberts, 2015; Ton et al., 2018): our relative error decreases with $m$, whereas the nonstationary RFF stays near $107\%$, which reflects the estimator's bias (Appendix A).

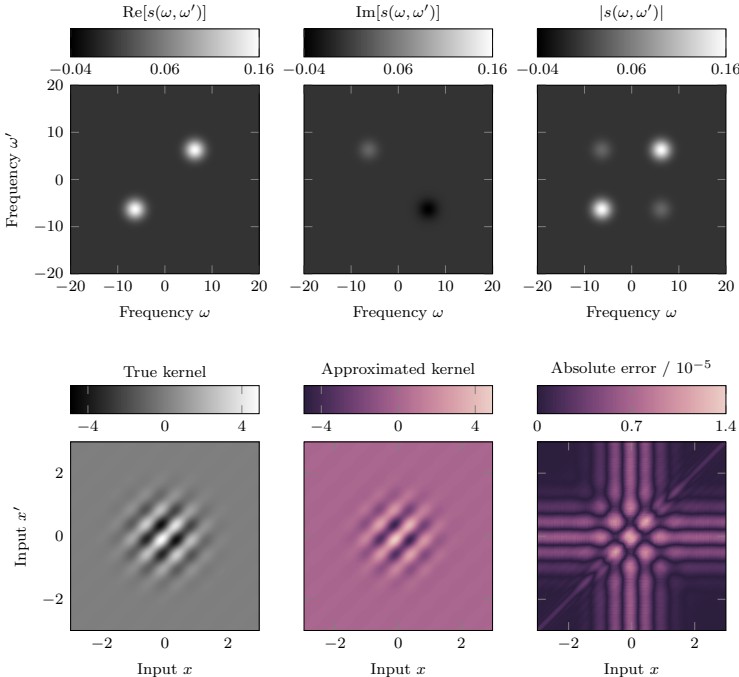

Figure 3: HMK approximation with complex-valued spectral density. Top row: real, imaginary, and absolute values of $s(\omega, \omega')$. Bottom row: true kernel, approximation, and absolute error.

**Harmonizable Mixture Kernel**  We consider an HMK (Shen et al., 2019). For a single component centered at the origin and no input scaling, the kernel is

$$k_{\text{HMK}}(x, x') = k_{\text{LSK}}(x, x') \sum_{i,j=1}^{Q} B_{ij} \exp\big(i(\eta_i x - \eta_j x')\big), \tag{29}$$

where $\eta_1, \ldots, \eta_Q$ are angular frequencies, $\boldsymbol{B} \in \mathbb{C}^{Q \times Q}$ is a positive semi-definite matrix, and $k_{\text{LSK}}$ is the kernel defined above. The spectral density is

$$s_{\text{HMK}}(\omega, \omega') = \sum_{i,j=1}^{Q} B_{ij}\, s_{\text{LSK}}(\omega - \eta_i, \omega' - \eta_j), \tag{30}$$

which can be complex-valued. We use $Q = 2$ with conjugate frequencies $\eta_1 = 2\pi$, $\eta_2 = -2\pi$ and a positive semi-definite matrix

$$\boldsymbol{B} = \begin{pmatrix} 2 & \frac{1}{2}i \\ -\frac{1}{2}i & 2 \end{pmatrix}, \tag{31}$$

producing a real-valued kernel with a complex-valued spectral density.

To approximate the single-component HMK, we use a symmetric frequency grid ($2m$ nodes, $4m$ rank) with $m = 100$ and cutoff frequency $\omega_m = 20$. The sample locations are $x_i = i\Delta x$ for $i = -\frac{n-1}{2}, \ldots, \frac{n-1}{2}$ with $\Delta x = 0.01$ and $n = 599$. Figure 3 shows the real, imaginary, and absolute values of the spectral density (top row) along with the true kernel, approximation, and absolute error (bottom row). The approximation accurately captures the complex oscillatory structure with absolute errors on the order of $10^{-5}$.

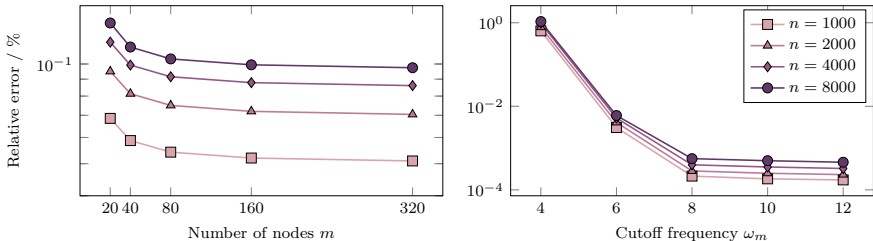

Figure 4: Ablation studies on the LSK showing relative error versus (left) number of nodes $m$ with fixed $\omega_m = 5$, and (right) cutoff frequency $\omega_m$ with fixed $m = 100$, across different problem scales $n$.

**Ablation Studies**   To investigate sensitivity to the number of nodes $m$ and the cutoff frequency $\omega_m$, we perform ablation studies on the LSK ($a = 1$) across different problem scales $n \in \{1000, 2000, 4000, 8000\}$ with $\Delta x = 0.001$. We measure the approximation quality with the relative error (26).

Figure 4 shows the results. The left panel examines error versus $m$ with fixed $\omega_m = 5$, while the right panel examines error versus cutoff frequency $\omega_m$ with fixed $m = 100$. Increasing $m$ consistently reduces approximation error across all problem scales, with diminishing returns beyond $m \approx 160$. For the cutoff frequency, errors decrease as $\omega_m$ increases from 4 to 6, capturing more spectral content, and plateau beyond $\omega_m \approx 8$. Larger problem scales show uniformly higher error at a fixed grid, but the overall trends remain consistent.

The value $\omega_m = 6$ observed in the ablation study corresponds to the point where the spectral density has decayed to negligible levels. For the LSK, we have $s_{\mathrm{LSK}}(6, 6) \approx 10^{-6}\, s_{\mathrm{LSK}}(0, 0)$, confirming that this cutoff captures most spectral mass. In practice, selecting $\omega_m$ is more critical than choosing $m$ (or $\Delta\omega$). We recommend (1) setting $\omega_m$ based on spectral content or the Nyquist limit, then (2) choosing $m$ large enough to satisfy the aliasing condition $\Delta\omega < \pi/x_{\max}$.

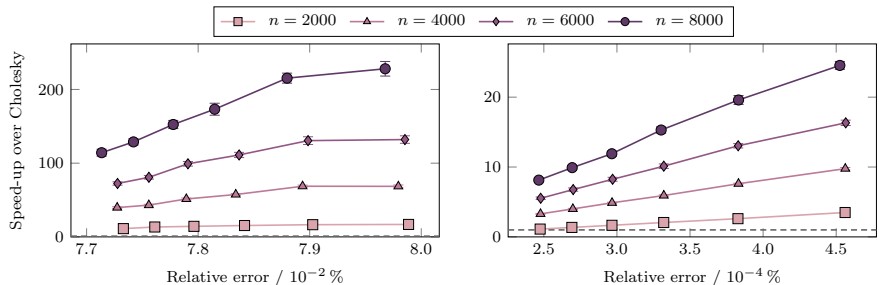

Figure 5: Speed-up of a single marginal log-likelihood evaluation over Cholesky, against the relative kernel error (26), for the LSK (left) and HMK (right) across problem scales $n$. Each curve has $m \in \{100, 120, \ldots, 200\}$, with error decreasing as $m$ grows. Speed-ups are means of 20 runs with 95 % confidence intervals. The dashed line marks equal execution times.

**Computational Efficiency**   We evaluate the computational efficiency of our approach against the standard dense Cholesky factorization. Both methods use a Gaussian likelihood with noise variance $\sigma_{\mathrm{noise}}^2 = 0.1$ and perform exact GP inference. They differ only in the kernel. The baseline factorizes the full matrix by a dense Cholesky, whereas ours uses the Woodbury identity (rank $m$ for the LSK and rank $4m$ for the HMK). We time a single marginal log-likelihood evaluation for both, varying $m \in \{100, 120, \ldots, 200\}$ across problem scales $n \in \{2000, 4000, 6000, 8000\}$.

Figure 5 plots the speed-up against the relative kernel error (26). The speed-up grows with the problem size $n$. At $n = 8000$ our method is up to $228\times$ (LSK) and $25\times$ (HMK) faster than Cholesky, at errors below $8 \times 10^{-2}\,\%$ and $5 \times 10^{-4}\,\%$. Although not clearly visible in the figure, for $n = 2000$ and $m = 200$ our HMK approximation remains faster than the dense Cholesky.

## 4.2 Kernel Learning

Whereas the approximation is our main contribution, we present kernel learning as a feasibility demonstration. We use a standard neural network as formulated in Section 3.3. We first address a reconstruction problem on a real-world dataset and then perform a small benchmark on synthetic datasets. All models are exact GPs with learnable noise variance and are trained with AMSGrad (Sashank J. Reddi et al., 2018) (learning rate $10^{-2}$, 2000 iterations).

To quantify our results, we use the relative error (26), negative log predictive density (NLPD), mean absolute error (MAE), and Kullback–Leibler (KL) divergence to the true posterior. We report the subset of metrics appropriate to each experiment.

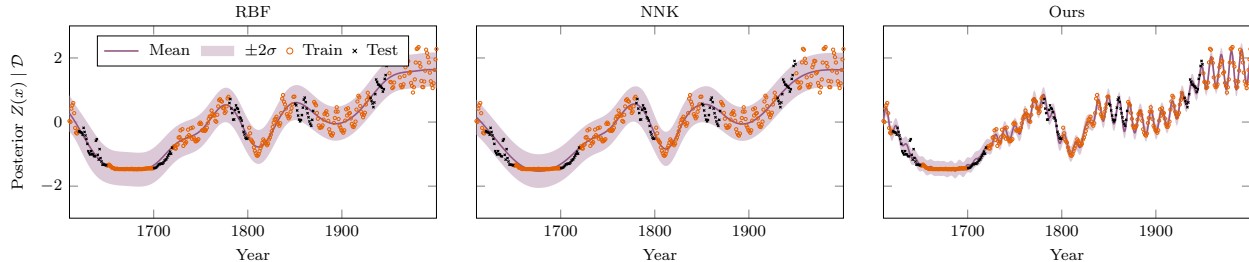

Figure 6: Posterior predictions on solar irradiance data with five held-out intervals. Metrics in parentheses are NLPD / MAE on the test data. Top: RBF $(-0.09 \,/\, 0.18)$. Center: NNK $(-0.07 \,/\, 0.20)$. Bottom: our method $(-0.43 \,/\, 0.13)$.

**Solar Irradiance Reconstruction**  We use the solar irradiance dataset (Lean, 2004) to demonstrate our approach on real-world data. In our model, we parametrize the spectral density using the factorized form (20) with a neural network having two hidden layers of 128 units and rank $r = 8$.

To mitigate the spectral bias of the neural network toward low frequencies, we first map the input frequency $\omega$ through a random harmonics embedding $\boldsymbol{g} \colon \mathbb{R} \to \mathbb{R}^{2q}$ (Tancik et al., 2020)

$$
\begin{aligned}
\boldsymbol{g}_{\sin}(\omega) &= \big(\sin(E_1 \,\omega/\omega_m), \dots, \sin(E_q \,\omega/\omega_m)\big), \\
\boldsymbol{g}_{\cos}(\omega) &= \big(\cos(E_1 \,\omega/\omega_m), \dots, \cos(E_q \,\omega/\omega_m)\big), \\
\boldsymbol{g}(\omega) &= \big(\boldsymbol{g}_{\sin}(\omega), \boldsymbol{g}_{\cos}(\omega)\big), \quad E_i \sim \mathcal{N}(0, \sigma_{\mathrm{emb}}^2),
\end{aligned}
\tag{32}
$$

where the $E_i$ are sampled once and held fixed. We set $q = 256$ and $\sigma_{\mathrm{emb}} = 15$.

The embedding is the input to the network $\boldsymbol{h} \colon \mathbb{R}^{2q} \to \mathbb{C}^r$, so that the feature map is the composition $\boldsymbol{f} = \boldsymbol{h} \circ \boldsymbol{g}$. We also include a learnable global scale parameter $\gamma^2$, which yields the model

$$
s(\omega, \omega') = \gamma^2 (\langle \boldsymbol{f}(\omega'), \boldsymbol{f}(\omega) \rangle + \langle \boldsymbol{f}(-\omega), \boldsymbol{f}(-\omega') \rangle).
\tag{33}
$$

We use a symmetric frequency grid with $m = 100$ and cutoff frequency $\omega_m = 32$. We place a zero-mean, unit-variance Gaussian prior on the network weights and obtain a maximum *a posteriori* (MAP) estimate.

We compare against two GP baselines: a radial basis function (RBF) kernel and a neural network kernel (NNK) (Williams, 1996). Figure 6 shows the posterior predictions. While RBF and NNK miss the high-frequency components of the data, our model recovers them through the random harmonics embedding; without this embedding, it underfits. However, this added flexibility can lead to overfitting, a known problem of overly flexible kernels (Ober et al., 2021).

**Benchmark with Synthetic Data**  We generate 20 synthetic datasets from the LSK (27) with $a = 0.5$ on $n = 35$ equally spaced training points in $-5 \le x \le 5$, and $t = 50$ test points drawn uniformly at random. The data follow $Z(x) = F(x) + E(x)$, where $F(x) \sim \mathcal{GP}(0, k_{\mathrm{LSK}}(x, x'))$ is a zero-mean GP and $E(x)$ is white Gaussian noise with $\sigma_{\mathrm{data}}^2 = 10^{-3}$. Similarly, we generate 20 datasets from the HMK (29) in $-2 \le x \le 2$

Table 3: Kernel learning on synthetic data (20 datasets each from the LSK and HMK). Metrics are relative error, NLPD, and KL divergence to the true posterior, expressed as the mean with $95\%$ confidence intervals. Best means in each column are in bold.

| | LSK | | | HMK | | |
| Method | Relative error / % | NLPD | KL | Relative error / % | NLPD | KL |
|---|---|---|---|---|---|---|
| RBF | $109.3 \pm 20.6$ | $-\mathbf{1.94} \pm 0.06$ | $7.5 \pm 1.8$ | $140.2 \pm 29.2$ | $0.12 \pm 0.41$ | $637.2 \pm 347.5$ |
| NNK | $350.7 \pm 213.3$ | $-\mathbf{1.94} \pm 0.06$ | $\mathbf{6.5} \pm 1.4$ | $100.0 \pm 0.0$ | $1.22 \pm 0.29$ | $3458.1 \pm 1432.8$ |
| NGSM | $165.9 \pm 82.6$ | $-1.78 \pm 0.14$ | $10.1 \pm 2.4$ | $\mathbf{65.8} \pm 13.3$ | $-0.34 \pm 0.19$ | $14.3 \pm 2.6$ |
| DKL | $913.7 \pm 537.1$ | $-0.59 \pm 0.30$ | $27.7 \pm 3.0$ | $286.8 \pm 44.4$ | $1.47 \pm 0.73$ | $38.2 \pm 4.3$ |
| Ours (real) | $\mathbf{77.9} \pm 15.9$ | $-1.93 \pm 0.06$ | $7.1 \pm 2.4$ | $68.6 \pm 9.6$ | $-\mathbf{0.55} \pm 0.11$ | $\mathbf{10.9} \pm 2.5$ |
| Ours (complex) | $82.3 \pm 13.5$ | $-1.84 \pm 0.10$ | $8.3 \pm 2.5$ | $78.7 \pm 7.7$ | $-0.54 \pm 0.12$ | $\mathbf{10.9} \pm 2.3$ |

using the same sampling scheme ($n = 35$ training, $t = 50$ test) with $\sigma_{\text{data}}^2 = 10^{-2}$. The HMK parameters are exactly those of Section 4.1.

The spectral density is again parametrized with the factorized form (20) using a neural network with two hidden layers of 128 units and rank $r = 8$, here without the random harmonics embedding. As in the solar experiment, we include a learnable global scale parameter $\gamma^2$ in the spectral density model.

Our model comes in two versions based on the network output: (1) real-valued $\boldsymbol{f} \colon \mathbb{R} \to \mathbb{R}^r$ and (2) complex-valued $\boldsymbol{f} \colon \mathbb{R} \to \mathbb{C}^r$. Both use a symmetric frequency grid with $m = 128$ and cutoff frequency $\omega_m = 10$, and we obtain a MAP estimate by placing a Gaussian prior on the network weights.

We compare against four baselines: the RBF kernel, the NNK (Williams, 1996), deep kernel learning (DKL) (Wilson et al., 2016), and the neural generalized spectral mixture (NGSM) kernel (Remes et al., 2018). The DKL feature network and the NGSM parameter network each use two hidden layers of 128 units, matching our model. NGSM uses two mixture components and a Gaussian prior on the network weights, as in the original work.

Table 3 reports test-data metrics for all baselines on both synthetic datasets (LSK and HMK). Our models achieve the best or comparable performance on most metrics. Interestingly, the real-valued version outperforms the complex-valued one on most metrics, even though the HMK data are generated from a complex-valued spectral density. This may stem from the greater optimization difficulty of the complex-valued model, or from an inductive bias of the real-valued model toward better solutions.

While our parametrization can match or improve on the baselines, training remains prone to underfitting without the harmonics embedding and to overfitting with it. Developing more robust optimization strategies is an important direction for future work.

## 5 Conclusion

We presented a method for constructing regular Fourier features for harmonizable Gaussian processes by discretizing the spectral representation on a regular frequency grid. Unlike existing RFF approaches for nonstationary kernels, our method yields a consistent approximation (Theorem 2) and does not require the spectral density to be a probability measure.

By factorizing the spectral matrix as $\boldsymbol{S}\Delta\omega^2 = \boldsymbol{C}\boldsymbol{C}^\dagger$, we obtain a low-rank kernel approximation that is positive semi-definite by construction and applies to kernels with arbitrary complex-valued spectral densities. This guarantee arises directly from the factorization structure rather than from modified Monte Carlo sampling. Moreover, the parametrization $s(\omega, \omega') = \langle \boldsymbol{f}(\omega'), \boldsymbol{f}(\omega) \rangle + \langle \boldsymbol{f}(-\omega), \boldsymbol{f}(-\omega') \rangle$ provides a principled framework for kernel learning via marginal-likelihood maximization or MAP estimation.

We demonstrated high approximation accuracy on the LSK and the HMK (complex-valued spectral density), the latter representing a unique capability unavailable to existing RFF methods. For kernel learning, we

reconstructed real-world solar irradiance data. On synthetic benchmarks, our approach matched or improved on competitive baselines.

The computational cost scales as $\mathcal{O}(nm^2)$ for approximation and $\mathcal{O}(nmr)$ for learning, potentially becoming prohibitive for very large datasets. For kernel learning, optimizing the general factorized form remains challenging, as the model is prone to underfitting or overfitting depending on its flexibility.

The method assumes finite spectral support, which reflects the reality of discrete sampling: the Nyquist theorem fundamentally limits recoverable frequencies. Additionally, our approach approximates a periodic repetition of the true kernel, requiring the domain to be small enough that successive periods do not overlap.

Future work could examine more general multi-dimensional settings, develop adaptive frequency selection schemes, and combine our approach with approximate inference methods.

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

## A    Bias of Nonstationary Random Fourier Features

Let $k(x, x')$ be a harmonizable kernel on $\mathbb{R} \times \mathbb{R}$ whose spectral measure $\mu(\mathrm{d}\omega, \mathrm{d}\omega')$ is assumed to be a positive finite measure (or, when normalized, a probability measure). Note that the kernel and its spectral measure are related by (Loève, 1948; Yaglom, 1987)

$$k(x, x') = \iint_{\mathbb{R}^2} \exp(\mathrm{i}(\omega x - \omega' x'))\, \mu(\mathrm{d}\omega, \mathrm{d}\omega').$$

The naïve Monte Carlo approximation of the integral is pointwise unbiased, but its finite-sample estimates are not positive semi-definite in general; see Eq. (9). Samo & Roberts (2015); Ton et al. (2018) suggested an alternative Monte Carlo approximation that guarantees positive semi-definiteness of the finite-sample estimates.

**Proposition.**    If the frequency pairs $(\omega, \omega')$ are sampled from the spectral measure of the target kernel (as in Samo & Roberts (2015); Ton et al. (2018)), the resulting positive semi-definite finite-sample estimates form a biased estimator of the kernel that becomes unbiased only when the kernel is stationary.

*Proof.* Since the kernel is Hermitian $(k(x, x') = \overline{k(x', x)})$ and $\mu(\mathrm{d}\omega, \mathrm{d}\omega')$ is a positive finite measure, the spectral measure is symmetric, $\mu(\mathrm{d}\omega, \mathrm{d}\omega') = \mu(\mathrm{d}\omega', \mathrm{d}\omega)$.

Sampling frequency pairs from $\mu$ and averaging four terms (Samo & Roberts, 2015; Ton et al., 2018)

$$k_{\mathrm{mod}}(x, x') = \frac{1}{4} \iint_{\mathbb{R}^2} \big(\exp(\mathrm{i}(\omega x - \omega' x')) + \exp(\mathrm{i}(\omega' x - \omega x'))$$
$$+ \exp(\mathrm{i}\omega(x - x')) + \exp(\mathrm{i}\omega'(x - x'))\big)\, \mu(\mathrm{d}\omega, \mathrm{d}\omega'),$$

is equivalent to replacing $\mu$ by the modified spectral measure

$$\mu_{\mathrm{mod}}(\mathrm{d}\omega, \mathrm{d}\omega') = \frac{1}{4} (\underbrace{\mu(\mathrm{d}\omega, \mathrm{d}\omega') + \mu(\mathrm{d}\omega', \mathrm{d}\omega)}_{=2\mu(\mathrm{d}\omega, \mathrm{d}\omega')}) + \frac{1}{4} (\underbrace{\delta_\omega(\mathrm{d}\omega')\mu_1(\mathrm{d}\omega) + \delta_{\omega'}(\mathrm{d}\omega)\mu_2(\mathrm{d}\omega')}_{=2\delta_\omega(\mathrm{d}\omega')\mu_1(\mathrm{d}\omega)}),$$

$$\mu_{\mathrm{mod}}(\mathrm{d}\omega, \mathrm{d}\omega') = \frac{1}{2}\mu(\mathrm{d}\omega, \mathrm{d}\omega') + \frac{1}{2}\delta_\omega(\mathrm{d}\omega')\mu_1(\mathrm{d}\omega),$$

with marginals $\mu_1(\mathrm{d}\omega) = \mu(\mathrm{d}\omega, \mathbb{R})$ and $\mu_2(\mathrm{d}\omega') = \mu(\mathbb{R}, \mathrm{d}\omega')$, and where $\delta_\omega(\cdot)$ denotes the point mass at $\omega$.

This modification yields a Monte Carlo approximation of the following kernel

$$k_{\mathrm{mod}}(x, x') = \frac{1}{2} \underbrace{\iint_{\mathbb{R}^2} \exp(\mathrm{i}(\omega x - \omega' x'))\, \mu(\mathrm{d}\omega, \mathrm{d}\omega')}_{k(x,x')} + \frac{1}{2} \iint_{\mathbb{R}^2} \exp(\mathrm{i}(\omega x - \omega' x'))\, \delta_\omega(\mathrm{d}\omega')\mu_1(\mathrm{d}\omega),$$

$$k_{\mathrm{mod}}(x, x') = \frac{1}{2} k(x, x') + \frac{1}{2} \underbrace{\int_{\mathbb{R}} \exp(\mathrm{i}\omega(x - x'))\, \mu_1(\mathrm{d}\omega)}_{k_{\mathrm{stat}}(x,x')},$$

$$k_{\mathrm{mod}}(x, x') = \frac{1}{2} k(x, x') + \frac{1}{2} k_{\mathrm{stat}}(x, x'),$$

which in general does not approximate $k(x, x')$. The bias vanishes only when $k(x, x') = k_{\mathrm{stat}}(x, x')$, in which case $k(x, x')$ itself is stationary. $\qquad\square$

## B    Error Analysis

The spectral representation (1) of harmonizable stochastic processes is a Riemann–Stieltjes integral in quadratic mean (q.m.), defined as (Loève, 1978, Sec. 37.3)

$$Z(x) = \int_{\mathbb{R}} \exp(\mathrm{i}\omega x)\, \mathrm{d}\Gamma(\omega) = \lim_{a,b\to\infty} \lim_{D\to 0} \sum_{k=1}^{n} \exp(\mathrm{i}\omega'_k x)\, (\Gamma(\omega_{k+1}) - \Gamma(\omega_k)),$$

with $D = \max_k(\omega_{k+1} - \omega_k)$, $-a = \omega_1 < \omega_2 < \ldots < \omega_{n+1} = b$, and any $\omega'_k \in [\omega_k, \omega_{k+1}]$. By the direct assertion of the harmonizability theorem (Loève, 1978, Sec. 37.4), the defining limit exists in q.m.; that is, the finite Riemann–Stieltjes sums converge to $Z(x)$.

Our method is the special case with equidistant spacing $D = \Delta\omega$, left endpoints $\omega'_k = \omega_k = k\Delta\omega$, and symmetric cutoff $a = b = \omega_m = m\Delta\omega$. Furthermore, our approach replaces the exact increment covariance by $\mathrm{E}[WW^\dagger] \approx \boldsymbol{S}\Delta\omega^2$ with $(\boldsymbol{S})_{ij} = s(\omega_i, \omega_j)$, made precise in Remark 3.

Our method thus makes two approximations, a finite cutoff $\omega_m$ and a nonzero spacing $\Delta\omega$, yielding the kernel approximation $\hat{k}(x, x')$ in (18). Throughout, $\mathcal{I}_{ij} := [\omega_i, \omega_{i+1}) \times [\omega_j, \omega_{j+1})$ denotes the $(i, j)$ grid cell. These cells cover the symmetric square $\mathcal{I}_m = [-\omega_m, \omega_m]^2$ of Section 3.

*Proof of Lemma 1.* We use Loève (1978, Sec. 37.2). Existence and finiteness of the second generalized derivative at every diagonal point are equivalent to q.m. differentiability of $\{\Gamma(\omega)\}$. The corollaries therein show that the mixed partial $s(\omega, \omega')$ exists and is finite on all of $\mathbb{R}^2$, not merely on the diagonal, and identify it as the covariance of the derivative process $\{\Gamma'(\omega)\}$. Since the covariance $s(\omega, \omega')$ is continuous at every diagonal point by assumption, it is continuous on $\mathbb{R} \times \mathbb{R}$. This continuity has two consequences. First, it yields $\mathrm{dd}'S(\omega, \omega') = s(\omega, \omega')\,\mathrm{d}\omega\,\mathrm{d}\omega'$; inserting this density into the bounded-variation property $\iint_{\mathbb{R}^2} |\mathrm{dd}'S(\omega, \omega')| = \iint_{\mathbb{R}^2} |s(\omega, \omega')|\,\mathrm{d}\omega\mathrm{d}\omega' \leq c < \infty$ shows that the spectral density is integrable. Second, $\{\Gamma'(\omega)\}$ is q.m. continuous. $\qquad\square$

*Proof of Theorem 2.*
*(i)* The approximation $\hat{k}(x, x')$ is the left-endpoint Riemann sum of the spectral integral (4) restricted to $\mathcal{I}_m$. Split the integral into the covered square and its complement, $k(x, x') = \iint_{\mathcal{I}_m} + \iint_{\mathbb{R}^2 \setminus \mathcal{I}_m}$. The truncation error is the contribution of the complement and is bounded as

$$\left| \iint_{\mathbb{R}^2 \setminus \mathcal{I}_m} \exp(\mathrm{i}(\omega x - \omega' x'))\, s(\omega, \omega')\,\mathrm{d}\omega\mathrm{d}\omega' \right| \leq \iint_{\mathbb{R}^2 \setminus \mathcal{I}_m} |s(\omega, \omega')|\,\mathrm{d}\omega\mathrm{d}\omega'.$$

For the covered square, write $g(\omega, \omega') := \exp(\mathrm{i}(\omega x - \omega' x'))\, s(\omega, \omega')$ and decompose the domain into the grid cells

$$\left| \hat{k}(x, x') - \iint_{\mathcal{I}_m} g(\omega, \omega')\,\mathrm{d}\omega\mathrm{d}\omega' \right| \leq \sum_{i,j=-m}^{m-1} \iint_{\mathcal{I}_{ij}} \left| g(\omega_i, \omega_j) - g(\omega, \omega') \right|\,\mathrm{d}\omega\mathrm{d}\omega'.$$

On each cell $\mathcal{I}_{ij}$, adding and subtracting $\exp(\mathrm{i}(\omega_i x - \omega_j x'))\, s(\omega, \omega')$ and using $|\mathrm{e}^{\mathrm{i}a} - \mathrm{e}^{\mathrm{i}b}| \leq |a - b|$ gives

$$
\begin{aligned}
\left| g(\omega_i, \omega_j) - g(\omega, \omega') \right| = \big| &\exp(\mathrm{i}(\omega_i x - \omega_j x')) \left( s(\omega_i, \omega_j) - s(\omega, \omega') \right) \\
&- (\exp(\mathrm{i}(\omega x - \omega' x')) - \exp(\mathrm{i}(\omega_i x - \omega_j x')))\, s(\omega, \omega') \big| \\
\leq\ & |s(\omega_i, \omega_j) - s(\omega, \omega')| + |x(\omega - \omega_i) - x'(\omega' - \omega_j)|\, |s(\omega, \omega')| \\
\leq\ & |s(\omega_i, \omega_j) - s(\omega, \omega')| + (|x| + |x'|)\Delta\omega\, |s(\omega, \omega')|
\end{aligned}
$$

and the oscillation of $s(\omega, \omega')$ over each cell is at most $\rho_m(\sqrt{2}\,\Delta\omega)$, since $s(\omega, \omega')$ is continuous, hence uniformly continuous, on the compact closure of $\mathcal{I}_m$. Integrating and summing over the $4m^2$ cells yields

$$\sum_{i,j=-m}^{m-1} \iint_{\mathcal{I}_{ij}} |g(\omega_i, \omega_j) - g(\omega, \omega')|\,\mathrm{d}\omega\mathrm{d}\omega' \leq 4\omega_m^2 \rho_m\big(\sqrt{2}\,\Delta\omega\big) + (|x| + |x'|)\,\Delta\omega\, c.$$

Combining the two parts, the discretization and the truncation error, completes the proof.

*(ii)* Let $\mathcal{X} \subset \mathbb{R}$ be bounded, and let $b < \infty$ be such that $|x| + |x'| \leq 2b$ for all $x, x' \in \mathcal{X}$. Then part (i) gives

$$|\hat{k}(x, x') - k(x, x')| \leq 4\omega_m^2 \rho_m(\sqrt{2}\,\Delta\omega) + 2b\,\Delta\omega\, c + \iint_{\mathbb{R}^2 \setminus \mathcal{I}_m} |s(\omega, \omega')|\,\mathrm{d}\omega\mathrm{d}\omega'.$$

At fixed cutoff $\omega_m$, the modulus $\rho_m(\cdot)$ is determined by $s(\omega, \omega')$ on the fixed compact closure of $\mathcal{I}_m$ alone and satisfies $\rho_m(\delta) \to 0$ as $\delta \to 0$. Hence,

$$4\omega_m^2 \rho_m(\sqrt{2}\,\Delta\omega) \to 0, \quad 2b\,\Delta\omega\,c \to 0,$$

as $\Delta\omega \to 0$. By the proof of part (i), these two terms also bound $\left|\hat{k}(x, x') - \iint_{\mathcal{I}_m} g(\omega, \omega')\,\mathrm{d}\omega\mathrm{d}\omega'\right|$, so $\hat{k}$ converges uniformly on $\mathcal{X} \times \mathcal{X}$ to this integral. In particular, the following limit exists and satisfies

$$\lim_{\Delta\omega \to 0} \sup_{x, x' \in \mathcal{X}} |\hat{k}(x, x') - k(x, x')| \leq \iint_{\mathbb{R}^2 \setminus \mathcal{I}_m} |s(\omega, \omega')|\,\mathrm{d}\omega\mathrm{d}\omega'.$$

Since $s(\omega, \omega')$ is integrable by Lemma 1 and $\mathcal{I}_m$ eventually contains every compact subset of $\mathbb{R}^2$, the truncation term vanishes as $\omega_m \to \infty$. $\qquad\square$

**Remark 3.** *Lemma 1 justifies the covariance structure underlying our model* $\mathrm{E}[WW^\dagger] \approx \boldsymbol{S}\Delta\omega^2$*, where $W$ collects the exact increments $W_k = \Gamma(\omega_k + \Delta\omega) - \Gamma(\omega_k)$. Since $\{\Gamma'(\omega)\}$ is q.m. continuous, the increments are recovered as q.m. integrals of the derivative process, $W_k = \int_{\omega_k}^{\omega_k + \Delta\omega} \Gamma'(\omega)\,\mathrm{d}\omega$, and taking second moments shows that the increment covariance equals the spectral mass of the grid cell*

$$\mathrm{E}[W_i \overline{W_j}] = \iint_{\mathcal{I}_{ij}} \mathrm{E}\big[\Gamma'(\omega)\overline{\Gamma'(\omega')}\big]\,\mathrm{d}\omega\,\mathrm{d}\omega' = \iint_{\mathcal{I}_{ij}} s(\omega, \omega')\,\mathrm{d}\omega\,\mathrm{d}\omega'.$$

*With $(\boldsymbol{S})_{ij} = s(\omega_i, \omega_j)$, our method replaces each cell mass by its left-endpoint value $\mathrm{E}[W_i \overline{W_j}] \approx s(\omega_i, \omega_j)\Delta\omega^2 = (\boldsymbol{S}\Delta\omega^2)_{ij}$, and $\hat{k}(x, x')$ is the corresponding left-endpoint Riemann sum. The kernel approximation error is therefore a deterministic quadrature error, which is why no stochastic argument appears in the proof of the error bound.*

## C    The Circular-Gaussian Assumption

**Proposition.**    Let $\{Z(x)\}$ be a zero-mean harmonizable GP with spectral distribution $S(\omega, \omega')$. Then there exists a zero-mean harmonizable GP $\{Y(x)\}$ with circular complex spectral increments that has the same spectral distribution $S(\omega, \omega')$, and hence the same kernel, as $\{Z(x)\}$.

*Proof.* Because $\{Z(x)\}$ is harmonizable, it has the spectral representation

$$Z(x) = \int_{\mathbb{R}} \exp(\mathrm{i}\omega x)\,\mathrm{d}V(\omega),$$

where $\{V(\omega)\}$ is a complex random measure with covariance $S(\omega, \omega') = \mathrm{E}[V(\omega)\,\overline{V(\omega')}]$.

There exists a circular Gaussian measure $\{\Gamma(\omega)\}$ with the properties (Loève, 1978, Sec. 37)

$$\mathrm{E}[\Gamma(\omega)\,\overline{\Gamma(\omega')}] = S(\omega, \omega'), \quad \mathrm{E}[\Gamma(\omega)\,\Gamma(\omega')] = 0.$$

We define a zero-mean harmonizable GP $Y(x) = \int_{\mathbb{R}} \exp(\mathrm{i}\omega x)\,\mathrm{d}\Gamma(\omega)$. Then $\{Y(x)\}$ has the kernel

$$\mathrm{E}[Y(x)\,\overline{Y(x')}] = \iint_{\mathbb{R}^2} \exp(\mathrm{i}(\omega x - \omega' x'))\,\mathrm{d}\mathrm{d}' S(\omega, \omega') = \mathrm{E}[Z(x)\,\overline{Z(x')}].$$

Hence $\{Z(x)\}$ and $\{Y(x)\}$ are both zero-mean Gaussian with the same kernel. Because $\{\Gamma(\omega)\}$ is circular Gaussian, its increments $W_i := \Gamma(\omega_i + \Delta\omega) - \Gamma(\omega_i)$ are circular Gaussian as well. Thus, for the approximation of the kernel, the circular-Gaussian assumption can be made without loss of generality. $\qquad\square$

