# OpenReview forum: "Regular Fourier Features for Nonstationary Gaussian Processes"
_TMLR — Under review for TMLR_

### Review · Reviewer_PL5j · 2026-06-24

**Summary Of Contributions:**

The paper proposes regular Fourier features for approximating and learning nonstationary Gaussian process kernels in the harmonizable-process setting. The main idea is to avoid interpreting the nonstationary spectral density $s(\omega,\omega')$ as a probability distribution, as is commonly done in random Fourier feature extensions. Instead, the paper directly discretizes the spectral representation on a regular frequency grid and preserves the covariance structure among the resulting spectral weights. This yields a finite-rank kernel approximation that is positive semi-definite by construction and is claimed to be consistent under a finite spectral support assumption.

The paper further introduces a factorized spectral parametrization that guarantees a valid nonstationary kernel and enables kernel learning from data. The method is demonstrated on locally stationary and harmonizable mixture kernels, including a case with a complex-valued spectral density, and is also evaluated on real and synthetic kernel-learning tasks.

A key strength of the paper is that it identifies and addresses a real limitation of existing nonstationary random Fourier feature methods, namely their reliance on a probability-measure interpretation of the spectral density. The resulting construction is mathematically clean and naturally ensures positive semi-definiteness. The ability to handle complex-valued spectral densities is also a useful contribution. However, the empirical evidence is somewhat limited, especially for the kernel-learning setting, and the paper would benefit from stronger error analysis, broader comparisons, and a clearer discussion of the method’s limitations in high-dimensional or non-product settings.

**Additional Comments:**

Overall, I found the paper interesting and technically well motivated. The proposed regular Fourier feature construction addresses a meaningful limitation of existing nonstationary random Fourier feature methods, and the positive semi-definite finite-rank formulation is appealing. The presentation is mostly clear, and the approximation experiments support the central idea.

My main reservations are that the theoretical guarantees are not yet formal enough for the strength of the claims, and the kernel-learning experiments remain somewhat preliminary. I would encourage the authors to sharpen the scope of the contribution, provide more precise approximation guarantees, and strengthen the empirical evaluation, especially for learned kernels and higher-dimensional settings. With these improvements, the paper could become a solid contribution to spectral methods for nonstationary Gaussian processes.

**Audience:**

Yes

**Audience Explanation:**

The paper addresses a real limitation of existing nonstationary Fourier feature methods: the need to interpret the nonstationary spectral density as a probability measure. By instead discretizing the harmonizable spectral representation directly and preserving correlations among spectral weights, the proposed method offers a conceptually clean way to obtain positive semi-definite finite-rank approximations for a broader class of nonstationary kernels. The ability to handle complex-valued spectral densities is also potentially useful for researchers interested in harmonizable kernels and nonstationary Gaussian processes.

**Claims And Evidence:**

No

**Claims Explanation:**

The main claims are partially supported, but not fully convincingly supported in their current form.

The paper provides clear derivations showing how discretizing the harmonizable spectral representation and factorizing the resulting spectral matrix yields a finite-rank kernel approximation that is positive semi-definite by construction. The approximation experiments on the locally stationary kernel and the harmonizable mixture kernel also provide useful evidence that the proposed method can accurately approximate known nonstationary kernels, including a case with a complex-valued spectral density.

However, some of the broader claims are not yet supported by sufficiently strong evidence. In particular, the paper repeatedly emphasizes consistency, but the theoretical support is mostly informal; there is no detailed error bound or convergence-rate analysis that quantifies the effects of spectral truncation, grid spacing, aliasing, domain size, and smoothness of the spectral density. The finite spectral support and periodicity assumptions are discussed, but their practical impact is not fully characterized.

The empirical evidence for kernel learning is also somewhat limited. The real-data experiment is restricted to a single solar irradiance example, and the synthetic benchmark uses small datasets with only modest and often statistically insignificant improvements over baselines. The authors themselves note that the method can underfit without the harmonic embedding and overfit with it, which suggests that the learning framework is still not very robust. In addition, the high-dimensional setting is only briefly addressed through product kernels, so the evidence does not fully support broad applicability to general multidimensional nonstationary kernels.

Overall, the central construction is sound and supported by reasonable evidence, but the stronger claims about consistency, practical kernel learning, and general applicability would require more complete theoretical analysis and stronger empirical validation.

**Requested Changes:**

Critical to securing my recommendation for acceptance:

1. Provide a more formal theoretical justification of the approximation claims. The paper repeatedly emphasizes consistency, but the current presentation lacks a precise convergence statement or error bound. I would like to see a theorem that characterizes the approximation error as a function of spectral truncation, frequency-grid spacing, domain size, and regularity of the spectral density. Even a relatively modest bound would make the main claim substantially more convincing.

2. Clarify the scope of the method in multidimensional settings. The paper briefly mentions product kernels as an extension, but this does not cover general multidimensional harmonizable kernels. The authors should clearly state whether the proposed method is intended mainly for one-dimensional or separable/product nonstationary kernels, and what computational or statistical barriers arise for general high-dimensional spectral densities.

3. Strengthen the empirical evidence for kernel learning. The current kernel-learning results are interesting but limited: the real-data evaluation uses one dataset, and the synthetic experiments show mostly comparable rather than clearly superior performance. Additional real datasets, larger sample sizes, more random seeds, and stronger runtime/memory comparisons would help support the claim that the method is practically useful beyond controlled approximation examples.

4. Better contextualize the contribution relative to deterministic Fourier features, quadrature-based kernel approximations, variational Fourier features, spectral mixture methods, and other low-rank kernel approximations. Since the use of regular spectral grids and Riemann-sum approximations is classical, the paper should more clearly delineate what is new: namely, the preservation of correlated spectral increments for harmonizable nonstationary processes and the resulting PSD finite-rank construction.

Changes that would strengthen the work but are not necessarily critical:

1. Add more detailed ablations for the learned model, including the rank $r$, cutoff frequency $\omega_m$, number of features $m$, use or removal of the random harmonics embedding, and the strength of the MAP prior. This would help readers understand when the method underfits or overfits.

2. Report optimization stability across random initializations. The paper notes that the learned model can underfit without the embedding and overfit with it, so it would be useful to quantify how sensitive the method is to initialization and hyperparameter choices.

3. Include wall-clock time and memory usage comparisons against the baselines. Since scalability is part of the motivation, empirical computational comparisons would make the practical contribution clearer.

4. Moderate some of the language around kernel learning. The approximation construction is the strongest part of the paper, while the learned-kernel component currently appears more preliminary. The claims should reflect this distinction more carefully.

---

> ### Author Response · Authors · 2026-07-17
> **Rebuttal to Review of Reviewer PL5j**
>
> We thank the reviewer for the constructive feedback. The revision addresses all four critical points: (1) a formal approximation-error theorem, (2) an explicit one-dimensional scope with the high-dimensional barriers stated, (3) a benchmark doubled to 20 datasets plus a wall-clock comparison, and (4) a rewritten related-work section that delineates our contribution.
>
> **1. Theoretical justification.**
> We added a theorem in the new subsection 3.2 `Approximation Error` (proofs in Appendix B). It bounds the approximation error in terms of the following quantities: the spectral truncation/cutoff w_m, the frequency-grid spacing dw, the domain size (the term `(|x|+|x'|) dw c` reflects the aliasing effect), and the regularity of s (modulus of continuity evaluated at the grid spacing). It proves consistency: the error vanishes uniformly on bounded domains as the cutoff grows and the spacing is refined.
>
> **2. Multidimensional scope.**
> The introduction now states explicitly that this work addresses one-dimensional settings. The `Assumptions and Limitations` paragraph (section 3.1) discusses the high-dimensional barrier. A general non-separable d-dimensional harmonizable kernel requires discretizing the joint spectral density on a 2d-dimensional grid, so storage and factorization grow exponentially in d and the computational gain is lost. Product kernels have O(ndm^2) cost but impose separability. We additionally note adaptive frequency-node placement as future work.
>
> **3. Empirical evidence for kernel learning.**
> We agree that additional real datasets and larger samples would further strengthen the evidence. In line with your non-critical point 4, the revision now presents kernel learning as a feasibility demonstration rather than a claim of empirical superiority. Nonetheless, we extended some results. (1) We doubled the synthetic benchmark to 20 datasets per kernel (Table 3). (2) For runtime, the revision adds a wall-clock-versus-accuracy comparison (section 4.1, `Computational Efficiency`). The timed operation, a single marginal-likelihood evaluation, corresponds to the per-iteration cost of training. For the learned model it is even cheaper than our low-rank approximation, since the factorized parametrization avoids the O(m^3) factorization of the spectral matrix.
>
> **4. Contextualization of the contribution.**
> The related-work discussion is now a dedicated section 2.2, extended and organized into three parts: (1) quadrature methods, (2) inference approximations (inducing points, variational Fourier features), and (3) parametric spectral models. We considered discussing classical low-rank factorizations of the kernel matrix (e.g., Nystrom), but these approximate the matrix of an already specified kernel, so they are orthogonal to spectral feature constructions. We therefore left them out. The parts contextualize our contribution: we apply the classical Riemann sum to the Fourier-Stieltjes integral of a harmonizable process and preserve the correlations among spectral increments, which yields the PSD finite-rank approximation.
>
> **Non-critical points.**
> (1, 2) We agree that ablations of the rank r, the harmonics embedding, the prior strength, and initialization stability would be informative. Under the feasibility framing we leave a systematic study of the learned model to future work. (3) Wall-clock and memory are addressed under point 3 above. (4) The abstract, contributions, and learning section now present the learning extension as a 'proof of concept' that 'matches competitive baselines', and the approximation as the primary, theory-backed contribution.
>
> **Further changes.**
> Beyond the requested changes, we resolved a notational ambiguity: m previously referred to several distinct quantities and is now uniformly defined by w_m = m dw. Section 3.1 now states the resulting ranks. For real-valued kernels with complex-valued spectral weights (e.g., the HMK), the low-rank approximation uses the symmetric grid of 2m nodes and has rank 4m. For real-valued spectral weights (e.g., the LSK), a one-sided grid of m nodes suffices and the rank is m.
>
> We hope these revisions resolve the reviewer's concerns, and we thank them again for feedback that has strengthened the paper.

---

> ### Comment · Reviewer_PL5j · 2026-07-18
>
> Thank you for the detailed response and substantial revision. The new version addresses my four main concerns: it provides a formal approximation-error result, clearly restricts the present method to the one-dimensional setting, strengthens the computational evaluation, and substantially improves the positioning relative to related spectral and quadrature methods. I also appreciate that kernel learning is now presented more appropriately as a proof-of-concept extension rather than as the primary empirically validated contribution.
>
> I consider the main concerns largely resolved. I have, however, two remaining technical points that I believe should be clarified or corrected.
>
> **1. Scope of the consistency statement.**
> Theorem 2 establishes the iterated limit
>
> $$
> \lim_{\omega_m\to\infty}\lim_{\Delta\omega\to 0}
> \widehat{k}(x,x')=k(x,x').
> $$
>
> The bound also gives uniform convergence over bounded input domains under this iterated limit. However, the statement in the rebuttal that the error vanishes “as the cutoff grows and the spacing is refined” may be read as convergence under an arbitrary joint limit. The current bound contains
>
> $$
> 4\omega_m^2\rho_m(\sqrt{2}\Delta\omega),
> $$
>
> where the modulus of continuity may itself depend on the expanding spectral domain. Thus, a joint limit would require an additional rate or global-regularity condition. Please qualify the claim as uniform convergence on bounded input domains under the stated iterated limit, or provide a sufficient condition for a joint limit. It would also be helpful to ensure that broad consistency claims are explicitly restricted to spectral densities satisfying the assumptions of Lemma 1.
>
> **2. Scaling and circularity in the real-valued construction.**
>
> With
>
> $$
> E=E_{\mathrm{re}}+iE_{\mathrm{im}},\qquad
> E_{\mathrm{re}},E_{\mathrm{im}}\sim
> \mathcal N(0,\tfrac12 I),
> $$
>
> one has
>
> $$
> \operatorname{Cov}\left(
> \operatorname{Re}[\phi(x)E],
> \operatorname{Re}[\phi(x')E]
> \right) =\frac12\operatorname{Re}\left[
> \phi(x)\phi(x')^\dagger
> \right].
> $$
>
> Equation (17), however, uses the expression without the factor (1/2). This can be fixed locally, for example by defining the real-valued process as
>
>
> $$
> \sqrt{2}\operatorname{Re}[\phi(x)E],
> $$
>
> or by changing the variance convention for the real and imaginary parts. Relatedly, circular complex weights and the Hermitian symmetry $W_{-k}=\overline{W_k}$ required for real-valued sample paths are not simultaneously immediate for the full symmetric frequency vector. Please clarify whether the real-valued subsection is a direct deterministic kernel construction or a stochastic process construction, and adjust the scaling accordingly.
>
> These points appear local and do not undermine the central Riemann-sum approximation theorem or the positive-semidefinite kernel construction. Subject to these clarifications, I regard my principal concerns as resolved and consider the revised paper a substantially stronger contribution.

---

> > ### Author Response · Authors · 2026-07-19
> > **Official Comment by Authors**
> >
> > We thank the reviewer for their updated assessment and the two observations. Both identified gaps, and we fixed them in the revision.
> >
> > **1. Scope of the consistency statement.**
> > The reviewer is right. The revision adopts the suggestions: (1) Theorem 2(ii) now states uniform consistency on bounded input domains explicitly as an iterated limit. (2) The modulus of continuity is now written rho_m to make its dependence on the spectral domain explicit. (3) The proof fixes w_m first (rho_m then refers to a fixed compact set) and shows that the inner limit exists via uniform convergence of k_hat to the truncated integral. We further note in the text that under the paper's finite-support assumption the truncation term vanishes and consistency already follows from the single limit dw -> 0 at fixed w_m, so no joint limit is invoked there. (4) The broad consistency claims are now explicitly restricted: the abstract states the mild-regularity requirement, and the contributions and the conclusion reference Theorem 2.
> >
> > **2. Scaling and circularity.**
> > The reviewer's observation is correct, and we thank them for catching it. The revised paragraph adopts the suggested scaling (sqrt(2) Re[alpha(x) W], Eq. (16)) and derives the factor from circularity (the real part carries half the power of the complex process). On the structural point: we sample circular W first and then impose Hermitian symmetry by symmetrizing the expansion, so Eq. (16) is the stochastic construction and its covariance yields the deterministic kernel approximation Eq. (17).

---

> > > ### Comment · Reviewer_PL5j · 2026-07-20
> > >
> > > Thank you for the detailed revisions. My concerns have been fully addressed, and I have no further questions.

---

### Review · Reviewer_26jc · 2026-06-25

**Summary Of Contributions:**

This paper proposes a new Gaussian process (GP) modelling within the theme of random Fourier features (RFF). It extends the standard RFF framework to non-stationary GPs (and learning), by combining numerical quadrature, low-rank approximation, and a spectral density factorization. The proposed method is new, and the method has been evaluated on synthetic and a solar irradiance reconstruction data.

# Strength

1. The paper is well presented. I have no difficulty understanding the method, and I do not find any technical errors. Also, to some extent, I think the paper is comprehensible also to audience outside of the GP community.
2. The proposed method allows for efficient computation in GP regression and learning indeed. (a small caveat: this has not been explicitly quantified). The empirical performance seems to be on par with baselines.

# Weakness

## Practical usability
- I cannot pin down a killer case where I really need to use the developed method. To address the computation problem of GP in general, there are
- The method focuses on $x \in R^d$ with $d=1$. This essentially limits the applicability of the method to scalar-input data only. It maybe extend to $d>1$, but it is not clear if this is feasible, and I don't think we retain the computational gain.
- To apply the spectral approximation with numerical quadrature, we need to know the band region of $\omega$ in advance. This means that for every new task, we have to do an exploratory data analysis (not clear how to do so) before applying this algorithm.

I think the limitaitons above make the method have limited usability in practice.

## Empirical results
I have some doubts with the experiments.

- The ultimate goal of RFF-based method is to solve the computation scaling problem of GPs. However, the paper does not have evidence for this. You should plot a figure where the x-axis is the time and the y-axis is the performance. In this way, we can confirm if the proposed method is indeed computationally efficient while being effective.
- None of the methods mentioned in Section 2.1 (also Table 1) was compared. I think at least one of the RFF-based methods for non-stationary processes should be compared.
- The empirical results are not significant. Most results focus on analysis of the method itself, while the comparison happens only in Table 3 and Figure 5. In Table 3, it is clear if the proposed method excel the baselines statistically, e.g., 10.7\pm2.9 vs 10.3\pm2.5. Also in Figure 5, why can't we just decrease the length scale of RBF kernel to get the periodic pattern (if this is intended)? Although TMLR guidelines do not expect the method to produce state-of-the-art results, the gain should be at least somewhat marginal, IMO.

**Additional Comments:**

- The $\dagger$ is not a commonly used notation for transpose/adjoint in ML.
- Equation (7), is it an inner product or outer product? The notation $\Phi \Phi^\dagger$ really reads like an outer product, but the result is an inner product
- Table 2. It is strange and is not expected that the "non-starionary RFF" stay around 100% relative error. This essentially says that this kernel is not working at all. What is the reason?
- Table 3. The relative errors are in average above 50% for all methods. This means that non of the methods give a valid approximation to the true covariance. Moreover, the NLPD and KL performance seem to be good enough regardless of the relative errors. Doesn't this imply "the approximation quality of the kernel does not affect the GP learning" and consequently weaken the contribution, (since I may not need a good enough approximation for real performance)?

**Audience:**

Yes

**Audience Explanation:**

The GP community would be interested in this type of construction.

**Broader Impact Concerns:**

No concerns.

**Claims And Evidence:**

Yes

**Claims Explanation:**

The claims were **mostly** supported by evidence, with a few minor points:

- "We demonstrate the method on harmonizable kernels, and apply it to kernel learning on real and synthetic data." This is not a contribution.  This is the evidence of the previous two bullet point contributions.

**Requested Changes:**

All my points in "Weakness" and "Additional comments".

---

> ### Author Response · Authors · 2026-07-17
> **Rebuttal to Review of Reviewer 26jc**
>
> We thank the reviewer for the constructive feedback. The revision adds an approximation-error theorem (section 3.2), a wall-clock-versus-accuracy comparison (section 4.1), an extended benchmark with 20 datasets (Table 3), and an explicit scope and limitations statement (section 3.1).
>
> **1. Practical usability.**
> The intended use case is band-limited data with nonstationary spectral structure. For example, surface texture is by definition band-limited (see ISO standards). Moreover, discrete time series always have a Nyquist limit as a natural spectral upper bound. Thus, this limit, known a priori, can serve as a default cutoff. For d > 1 we now state the limitation explicitly (`Assumptions and Limitations`, section 3.1). The one-dimensional setting is a constraint of the present paper. A general solution is left to future work.
>
> **2. Time versus performance.**
> The revision adds the requested evidence (section 4.1, `Computational Efficiency`): we time a single marginal-likelihood evaluation and plot the speed-up over a dense Cholesky against the relative kernel error (Figure 5). The timed operation corresponds to the per-iteration cost of training. For the learned model it is even cheaper than our low-rank approximation, since the factorized parametrization avoids the O(m^3) factorization of the spectral matrix.
>
> **3. Comparison to nonstationary RFF, and Table 2's 100 % error.**
> Table 2 compares against the nonstationary RFF of Samo & Roberts and Ton et al. Its 107 % error is expected, and it is our motivation: the modification that makes these estimators positive semi-definite is biased for every nonstationary kernel. It converges to (k + k_stat)/2, a mixture with a stationary kernel (see Appendix A). So the error persists no matter how many frequencies are sampled, as Table 2 shows across m. The revision now clarifies Table 2 with 'Comparison of our method against the nonstationary RFF (Samo & Roberts (2015), Ton et al. (2018)).' Furthermore, we note beside the table that this plateau reflects the estimator's bias.
>
> **4. Significance, and the relative errors in Table 3.**
> We doubled the benchmark to 20 datasets per kernel, and the revision presents kernel learning as a feasibility demonstration: we claim matching, not surpassing, the baselines. The relative errors above 50 % are not evidence that the methods are failing. Table 3 asks how well the data-generating kernel is recovered from n = 35 noisy observations. This is an ill-posed inverse problem in which many kernels explain the data comparably well. Predictive metrics are therefore robust to moderate kernel error, but large errors do show: on the HMK, the three largest kernel errors (DKL, RBF, NNK) are also the three worst NLPDs and KLs.
>
> **5. Figure 5 (now Figure 6): why not a shorter RBF length scale?**
> A shorter length scale adds small-scale variability but cannot reproduce the quasi-periodic structure in the held-out intervals. Within a gap wider than the length scale, the RBF posterior reverts to the prior mean with inflated variance, and shortening it accelerates this reversion (worsening NLPD). The length scale was set by marginal-likelihood optimization, so a shorter one is not favored by the data.
>
> **6. Notation and other comments.**
> We added the note 'Here (.)^† denotes the conjugate transpose' at its first use. For Eq. (7), we agree the Gram form was written ambiguously: scalar kernel expressions are now written as inner products <phi(x), phi(x')> with a stated conjugate-second convention, alpha(x) is declared a row vector, and the dagger/transpose are reserved for matrices. We also resolved an ambiguity in m: it is now uniformly defined as w_m = m dw, with the resulting ranks stated in section 3.1. Finally, as you suggested, we removed the third contribution bullet.
>
> We hope these revisions resolve the concerns, and we thank the reviewer again for feedback that has strengthened the paper.

---

### Review · Reviewer_36Kg · 2026-07-06

**Summary Of Contributions:**

## **Summary:**

The paper proposes **regular Fourier features for harmonizable, nonstationary Gaussian processes**. Its central idea is to avoid treating the two-frequency spectral density $s(\omega,\omega')$ as a probability density, which is generally inappropriate for nonstationary processes. Instead, the paper directly discretizes the harmonizable spectral representation on a regular frequency grid and preserves the covariance structure among spectral increments. By factorizing the discretized spectral covariance matrix $S\Delta\omega^2 = CC^\dagger$, the method obtains a finite-rank kernel approximation of the form $\phi(x)\phi(x')^\dagger$, which is positive semi-definite by construction.

The paper also extends this idea to kernel learning by parameterizing the spectral density through a factorized form,
$$
s(\omega,\omega') = f(\omega)^\dagger f(\omega') + f(-\omega')^\dagger f(-\omega),
$$
which guarantees a valid nonstationary spectral density and allows the kernel to be learned from data using marginal likelihood or MAP estimation.

## **Strength:**

- The paper clearly identifies a limitation of probability-based nonstationary Fourier feature methods and proposes a conceptually clean alternative that can handle complex-valued harmonizable spectral densities.

- The approximation experiments on locally stationary and harmonizable mixture kernels are convincing and demonstrate the benefit of preserving spectral-weight correlations.

- The resulting approximation is finite-rank and PSD by construction.

## **Weaknesses:**
- The method relies on finite spectral support and regular-grid discretization, which may limit scalability to high-dimensional inputs.

- The kernel-learning experiments are less conclusive than the approximation experiments, with modest empirical gains and some optimization difficulties.

- The related work discussion could also more clearly distinguish this method from prior parametric nonstationary spectral kernels, especially Remes et al. and Shen et al.

**Audience:**

Yes

**Audience Explanation:**

The paper should interest a subset of the TMLR audience, especially researchers working on Gaussian processes, spectral kernels, scalable kernel approximations, and nonstationary covariance modeling.

**Broader Impact Concerns:**

No major broader impact concerns.

**Claims And Evidence:**

No

**Claims Explanation:**

Some claims are less fully supported. The kernel-learning extension is demonstrated on real and synthetic datasets, but the empirical gains are modest and often not statistically significant. The authors themselves note that the learned model may underfit without the harmonics embedding and overfit with it, suggesting that the optimization and model-selection aspects are not yet fully resolved.

In addition, the paper would benefit from clearer evidence regarding the effects of finite spectral support, truncation, aliasing, and frequency-grid selection. These assumptions are discussed, but no formal error bounds or systematic sensitivity analysis beyond the provided ablations are given.

The high-dimensional case is also only briefly addressed through product kernels, which imposes separability and does not fully support claims of broad applicability to general multidimensional nonstationary kernels.

**Requested Changes:**

1. **Clarify the scope of Table 1 and related work.**
   The paper should more clearly distinguish Fourier-feature approximation methods from parametric nonstationary spectral kernel families. In particular, Remes et al. and Shen et al. are only briefly discussed (and there could be other recent references), although Shen et al.’s harmonizable mixture kernel is directly relevant and is used in the experiments. Either Table 1 should be restricted in scope, e.g. “Fourier-feature approximation methods,” or these related methods should be discussed more explicitly.

2. **Clarify assumptions and limitations.**
   The finite spectral support assumption, periodic aliasing effect, and frequency-grid selection criteria should be stated more prominently. The current discussion is useful, but the practical implications for approximation quality and failure modes should be clearer.

3. **Temper claims about kernel learning.**
   The approximation results are convincing, but the kernel-learning experiments are less conclusive. The paper should present the learning extension more as a promising feasibility demonstration rather than as strong empirical superiority over existing kernel-learning methods.

4. **Add more systematic sensitivity analysis.**
   The ablations on the number of features and cutoff frequency are helpful, but further analysis of truncation error, aliasing, and grid spacing would strengthen the empirical support.

5. **Discuss high-dimensional limitations more explicitly.**
   The product-kernel extension avoids the full multidimensional spectral grid but imposes separability. The paper should clarify that general high-dimensional harmonizable kernels remain challenging.

---

> ### Author Response · Authors · 2026-07-17
> **Rebuttal to Review of Reviewer 36Kg**
>
> We thank the reviewer for the constructive feedback. The revision adds (1) a formal approximation-error theorem, (2) a rewritten related-work section that delineates our contribution, (3) a consistent reframing of kernel learning as a feasibility demonstration, and (4) an explicit treatment of the high-dimensional limitations.
>
> **1. Scope of Table 1 and related work.**
> We have made both changes you suggested. Table 1 is now titled "Comparison of Fourier-feature approximation methods [...]". The related-work discussion is now a dedicated section 2.2, extended and organized into three parts: (1) quadrature methods, (2) inference approximations (inducing points, variational Fourier features), and (3) parametric spectral models. In (3) we now discuss Remes et al. (2017, 2018), Shen et al. (2019), Wilson and Adams (2013), and Benton et al. (2019) explicitly.
>
> **2. Assumptions and Limitations.**
> We renamed the `Assumptions` paragraph to `Assumptions and Limitations` in section 3.1. This paragraph now states each assumption together with its practical implication and failure mode: the finite-spectral-support assumption (truncation assumption) that is motivated by the Nyquist limit (w_m < pi/dx); the periodicity of the approximation, with dw chosen so that x_max < pi/dw and the resulting period-interference failure mode; and the diagonal jitter required for the Cholesky factorization, whose error is confined to the modeled band. This gives the reader the criteria needed to choose the grid and cutoff in advance.
>
> **3. Tempering claims about kernel learning.**
> We agree that the approximation is the stronger contribution and have reframed the learning results accordingly. The paper now states once, up front, that kernel learning is a "proof of concept" and a "feasibility demonstration," and that it "matches competitive baselines" rather than surpassing them. The abstract, the contribution list, and the learning section are all aligned to this framing.
>
> **4. Systematic sensitivity analysis.**
> Our new approximation-error theorem (section 3.2) provides the analytic characterization requested. It bounds the error as a function of the truncation w_m, the grid spacing dw, the domain size (the term `(|x|+|x'|) dw c` reflects the aliasing effect), and the regularity of the spectral density (modulus of continuity), and it proves consistency as these quantities are refined. The existing ablation studies cover those quantities empirically: at fixed cutoff w_m, sweeping m is a sweep of the grid spacing dw = w_m/m, and at fixed m, sweeping w_m changes the truncation error. Together with the `Assumptions and Limitations` paragraph, the theorem and the two ablation studies now give both analytic and empirical coverage. To address the scalability point raised, the revision also adds a wall-clock-versus-accuracy comparison on the LSK and HMK.
>
> **5. High-dimensional limitation.**
> The paragraph `Assumptions and Limitations` now discusses the high-dimensional barrier. A general non-separable d-dimensional harmonizable kernel requires discretizing the joint spectral density on a 2d-dimensional grid, so storage and factorization grow exponentially in d and the computational gain is lost. Product kernels have O(ndm^2) cost but impose separability. We position the contribution around one-dimensional problems and note adaptive frequency-node placement as future work.
>
> **Further changes.**
> Beyond the requested changes, we resolved a notational ambiguity: m previously referred to several distinct quantities and is now uniformly defined by w_m = m dw. Section 3.1 now states the resulting ranks. For real-valued kernels with complex-valued spectral weights (e.g., the HMK), the low-rank approximation uses the symmetric grid of 2m nodes and has rank 4m. For real-valued spectral weights (e.g., the LSK), a one-sided grid of m nodes suffices and the rank is m.
>
> We hope these revisions resolve the reviewer's concerns, and we thank them again for feedback that has strengthened the paper.